# Rapid and Widespread Distribution of Intranasal Small Extracellular Vesicles Derived from Mesenchymal Stem Cells throughout the Brain Potentially via the Perivascular Pathway

**DOI:** 10.3390/pharmaceutics15112578

**Published:** 2023-11-03

**Authors:** Weiwei Shen, Tongyao You, Wenqing Xu, Yanan Xie, Yingzhe Wang, Mei Cui

**Affiliations:** Department of Neurology, Huashan Hospital, Fudan University, Shanghai 200031, China; 20211220030@fudan.edu.cn (W.S.); tyyou15@fudan.edu.cn (T.Y.); 21211220027@m.fudan.edu.cn (W.X.); ynxie16@fudan.edu.cn (Y.X.); jnzhe@163.com (Y.W.)

**Keywords:** extracellular vesicles, intranasal administration, distribution, perivascular space

## Abstract

Intranasal administration is a promising strategy to enhance the delivery of the sEVsomes-based drug delivery system to the central nervous system (CNS). This study aimed to explore central distributive characteristics of mesenchymal stem cell-derived small extracellular vesicles (MSC-sEVs) and underlying pathways. Here, we observed that intranasal MSC-sEVs were rapidly distributed to various brain regions, especially in the subcortex distant from the olfactory bulb, and were absorbed by multiple cells residing in these regions. We captured earlier transportation of intranasal MSC-sEVs into the perivascular space and found an increase in cerebrospinal fluid influx after intranasal administration, particularly in subcortical structures of anterior brain regions where intranasal sEVs were distributed more significantly. These results suggest that the perivascular pathway may underlie the rapid and widespread central delivery kinetics of intranasal MSC-sEVs and support the potential of the intranasal route to deliver MSC-sEVs to the brain for CNS therapy.

## 1. Introduction

Globally, the burden of deaths and disabilities caused by neurological disorders is increasing and spurring new drug development [1]. The blood–brain barrier (BBB) is always the biggest obstacle for drugs to achieve an effective concentration in the lesions of CNS [2]. To improve efficacy and decrease the risk of systemic toxicity, solutions are needed to optimize the central delivery of the drugs [3]. The use of nanodrug delivery systems has demonstrated early preclinical success in pharmaceutical agents, including antibiotics, antineoplastic agents, and other therapeutic molecules through BBB for CNS therapy [4].

Similar in size to synthetic nanoparticles (<200 nm in diameter), small extracellular vesicles (sEVs), mainly exosomes and small microvesicles, have recently drawn the attention of drug developers as natural drug delivery carriers [5]. sEVs, secreted by almost all living cells, carry a great number of proteins and nucleic acid from donor cells in the lumen bounded by lipid bilayers. Emerging evidence has shown that sEVs can shuttle through the BBB via paracellular or transcellular pathways and deliver biotherapeutic agents to modulate the function of recipient cells [6]. In addition, sEVs have lower immunogenicity and less toxicity than existing drug carriers [7]. Efforts have been made in sEVs-based drug delivery systems loading either biologic (e.g., growth factors, cytokines, siRNAs, and miRNA) or small molecules (e.g., dopamine, curcumin, and doxorubicin) for the treatment of various CNS diseases from neurodegenerative diseases to neuroinflammatory diseases and from brain tumors to brain viral infections [8]. The strategies for active molecules loaded onto or into sEVs involving genetic engineering technology, drug coincubation, encapsulation, and surface modification are always updating to increase the bioavailability of drugs [9].

However, intravenously injected sEVs are mainly distributed to the liver and spleen, and they are easily cleared from the systemic circulation by macrophages [10,11,12]. The compromised levels of sEVs reaching the brain via widely used intravenous administration may be one of the crucial reasons for the unsatisfying efficacy of sEVs-based therapy for CNS diseases in clinical trials.

Intranasal delivery is a noninvasive and direct route from the nose to the brain, which largely bypasses BBB and the first-pass metabolism [13]. In recent years, there has been a surge in studies exploring the combination of sEVs treatment with intranasal administration in various fields of CNS disorders and harvesting promising therapeutic effects [14,15,16,17,18,19,20,21]. The rapid central enrichment of intranasal sEVs has been suggested to occur via direct intraneuronal transport along the axon of the olfactory or trigeminal nerves [22]. Several research works demonstrated the arrival of intranasal sEVs in the olfactory bulb (OB) and pons (Pn), the original brain entry sites, plus some distant brain areas, such as the striatum and frontal cortex [16,23], but clear evidence of distributive characteristics of intranasal sEVs in the brain has yet to be provided. The potential pathway behind distribution has also been unanswered. Previous studies have indicated that intranasal substances can cross the epithelium barrier and enter the cerebrospinal fluid (CSF) in the subarachnoid space via paracellular pathways when epithelium renewal starts [24]. There was a widespread distribution of intranasal macromolecules in the brain that mediated the exchange of CSF and interstitial fluid (ISF) within the perivascular space (PVS) of cerebral vessels [25,26]. The PVS of the brain, namely Virchow–Robin spaces, comprised between astrocyte endfeet and blood vessel walls, surrounds penetrating arteries, arterioles, and capillaries. PVS allows the substances within the CSF to be driven into the interstitial space as the exchange of CSF and ISF [27].

Therefore, we speculated that the perivascular pathway may underlie the distributive pattern of intranasal sEVs in the brain. In this study, we aimed to investigate the distributive characteristics and potential pathways of mesenchymal stem cell-derived sEVs (MSC-sEVs), the most commonly used therapeutic sEVs, within the brain following intranasal administration. And, we expected to provide new insights into the potential of intranasal MSC-sEVs-based therapy.

## 2. Materials and Methods

### 2.1. Isolation and Characterization of MSC-sEVs

MSCs derived from human umbilical cords (huc MSC) acquired from National Stem Cell Translational Resource Center were cultured in a special serum-free medium provided by the center and incubated at 37 °C in a humidified atmosphere with 5% CO_2_. When the confluency of cells was almost 100% in a T75 flask full of 10 mL medium, sEVs in the supernatants were isolated using a previously described protocol [28]. Briefly, the supernatants in each tube were sequentially centrifugated at 2000× *g* for 10 min and 10,000× *g* for 30 min at 4 °C to remove cell debris and microparticles. Parts of supernatants would be stored at −80 °C but used within 2 weeks. Around 30 mL supernatants in each tube followed a second ultracentrifugation at 120,000× *g* for 90 min at 4 °C in a Type Ti70 rotor using an L-80XP ultracentrifuge (Beckman Coulter, Brea, CA, USA). The pellets were then resuspended in phosphate-buffered saline (PBS) for immediate use.

The morphology of isolated sEVs was assessed by transmission electron microscopy (TEM), supported by the Chinese Academy of Sciences. Specific procedures will be described in Section 2.8. The concentration and size distribution of the isolated sEVs were evaluated using nanoparticle tracking analysis (NTA) at Viva Cell Shanghai with ZetaView PMX 110 (Particle Metrix, Meerbusch, Germany) and corresponding software ZetaView 8.04.02.

Western blot was carried out to detect EV markers. After extracting the whole proteins, the concentration was measured using a BCA assay kit (Sangon Biotech, Shanghai, China). An equivalent of 20 μg proteins were separated by 7.5% SDS-PAGE and then transferred onto PVDF membranes. After blocking with 5% skim milk, the membranes were incubated overnight at 4 °C with the following primary antibodies: anti-Alix (1:1000, Abcam 125011), anti-TSG101 (1:1000, Abcam 117600), and anti-CD63 (1:1000, Abcam 271286). The membrane was washed with 1× TBST 3 times for 10 min and incubated with secondary antibodies conjugated with horseradish peroxidase (HRP) for 1 h at room temperature (RT). Immunoreactivity was visualized with ChemiDoc™ XRS+ System (Bio-Rad Co., Hercules, CA, USA).

### 2.2. Labeling of MSC-sEVs

We referred to the procedures reported by Tian et al. for labeling the amino groups on sEVsomal proteins or phosphatidylethanolamine, instead of traditional labeling methods using lipophilic dyes to mark the membranes of sEVs (Appendix A) [29]. Specific procedures and concentrations of reactants have been altered a bit according to our practice. Briefly, 1 μL dibenzocyclooctyne-sulfo-N-hydroxysuccinimidyl ester (DBCO-sulfo-NHS,18 mM, Sigma, St. Louis, MO, USA) was added to sEVs in PBS (1 μg/μL, 200 μL) to a final concentration about 100 μM and allowed to react on a rotating shaker at RT for 3 h. Unconjugated DBCO-sulfo-NHS was removed by ultracentrifugation at 120,000× *g* for 90 min at 4 °C and sEVs were resuspended in PBS. Subsequently, 0.2 μL Cy3- or C7- azide (10 mM, Lumiprobe Co., Cockeysville, MD, USA) was added to a final concentration of about 10 μM and triggered a reaction of copper-free click chemistry on a rotating mixer at 4 °C overnight. The pellets were then floated on a 30% sucrose cushion and centrifuged at 120,000× *g* for 90 min at 4 °C to remove extra dyes lying at the interface between sucrose and PBS (Appendix A). According to the wavelength range, Cy7 and Cy3 labeled-sEVs (Cy7-sEVs, Cy3-sEVs) were exploited for near-infrared fluorescence (NIRF) imaging and immunofluorescence imaging, respectively.

To locate intranasal MSC-sEVs in brain tissue more precisely, we attempted to apply the method of labeling sEVs with glucose-coated gold nanoparticles (GNPs) mediated by the glucose transporter and endocytic proteins for TEM imaging [30]. Briefly, we incubated sEVs (1 μg/μL) with GNPs with a diameter of 5 nm (0.05 mg/mL, 5 mL, Xi’an Ruixi Biological Technology Co., Xi’an, China) at 37 °C for 3 h. After incubation, the sEVs were centrifuged at 120,000× *g* for 90 min at 4 °C to remove extra GNPs. The pellet was then resuspended in PBS and determined using TEM imaging (Appendix A).

### 2.3. Animals

C57BL/6J male mice (10–12 w of age) were purchased from Shanghai Jiesijie Laboratory Animal Co. and housed in the Experimental Animal Center of Fudan University in a temperature- and humidity-controlled specific-pathogen-free laboratory with a 12 h/12 h light/dark cycle. All procedures were performed by the Guide of the National Science Council of the People’s Republic of China. This study was approved by the Ethics Committee of Fudan University, Shanghai, China; IRB approval number: 2022JSHuashanHospital-283.

### 2.4. Intranasal Administration of MSC-sEVs

The control group and treated group, respectively, received PBS and huc MSC sEVs intranasally. Mice were anesthetized with ketamine: xylazine: PBS at a ratio of 1:0.5:8.5 by a single intraperitoneal (i.p.) injection in a final volume of 100 µL per 10 g mouse. Referring to the research conducted by Ullah et al., each animal was set on our original platform to maintain the head’s natural down-and-forward position (around 45° tilting) for better nasal absorption [31]. Both nostrils of mice were, respectively, treated with 10 µL of hyaluronidase (100 U, Sangon Biotech Co., Shanghai, China) in sterile PBS solution to enhance the permeability of the nasal mucous membrane. A total of 30 min later, each mouse was treated intranasally with 1 μg/μL MSC-sEVs (nearly 1 × 10^10^ particles/μL) or pure PBS in a volume of 20 µL. Specifically, we placed a ~2 µL drop by 10 µL micropipette very close to one nostril so that the mouse could directly inhale the droplet, and then we repeated the step with the other nostril 2 min later until the 20 µL dose was finished.

### 2.5. NIRF Imaging and Fluorescence Imaging

At 0.5, 2, 6, and 24 h after intranasal administration of Cy7-MSC-sEVs, mice in both groups (n = 4/group) were anesthetized by isoflurane (5% for induction, 2% for maintenance). The head of the mouse was shaved for imaging. At 2 h post administration, the tissues of the mouse, including brain, heart, lung, spleen, kidney, liver, and skull, were obtained for further imaging. A VISQUE In Vivo Smart-LF system (Vieworks Co., Anyang, Korea) with corresponding software CleVue, Version 3.1.3.2054 was employed to capture NIRF images.

At 2 h after intranasal administration of Cy3-MSC-sEVs, animals in both groups were anesthetized (i.p., ketamine: xylazine: PBS = 1:0.5:8.5) and perfused transcardially with 0.1 M PBS followed by 4% paraformaldehyde. A total of 100 µm-thick brain slices were cut using a vibratome (Leica Biosystems, Nussloch, Germany). The slices were imaged using Olympus Slideview VS120 and the software OlyVIA, Version 2.9.1.

### 2.6. Immunofluorescence Staining and Confocal Imaging

At 2 h post intranasal delivery, the brains and skulls of mice were dissected, postfixed in 4% paraformaldehyde overnight after anesthetized, and perfused transcardially. The brains were continuously dehydrated in 20% and 30% sucrose and frozen in liquid nitrogen after being embedded in Tissue-Tek^®^ O.C.T. Compound. A total of 30 µm-thick sagittal sections were cut using a cryostat (Leica) and collected serially. Several sets of serial sections were processed for immunofluorescence staining using primary antibodies against Iba-1 (abcam178846, 1:2000), GFAP (abcam4674, 1:1000), NeuN (abcam177487, 1:1000), and PDGFR-β (abcam32570, 1:200) and transported through aquaporin 4 (AQP4, Servicebio Co., Wuhan, China, 1:1000). And, the dura mater was stained by primary antibodies against lymphatic vessel endothelial receptor 1 (LYVE1, Cell signaling technology, Danvers, MA, USA, 1:200). The matching secondary antibodies conjugated to fluorescent probes were purchased from Thermo Fisher Scientific (Waltham, MA, USA) and were employed. Optical Z-sections were sampled from different brain regions using a Nikon C2Si confocal microscope (Tokyo, Japan) and relevant software NIS Elements, Version 5.01. Interested regions for sEVs incorporation quantitative analysis by astrocytes, microglia, and neurons was performed (n = 3), including the olfactory bulb (OB), cortex (Ctx), corpus callosum (CC), and other structures of subcortex (SCtx) selected, such as striatum (Str), hippocampus (Hip), pons (Pn), and cerebellum (Cb). To observe the location of sEVs with regard to PVS, more attention was paid to the CC, Hip, and Str where abundant perforating vessels are distributed.

### 2.7. TEM

Naïve MSC-sEVs or GNP-labeled MSC-sEVs (GNP-MSC-sEVs) were purified and resuspended in sterile water, and then were blotted onto carbon-coated 200-mesh copper electron microscopy grids for 5 min. Subsequently, sEVs were incubated with phosphotungstic acid (2%, 3 min, RT). Micrographs were obtained under a TEM (JEOL 1230, Tokyo, Japan) with the software Digital Micrograph (TM) Version 1.85.1535 at the Chinese Academy of Sciences.

At 1 h after intranasal delivery of GNP-MSC-sEVs, mice were anesthetized (i.p., ketamine: xylazine: PBS = 1:0.5:8.5). Intracardiac perfusion with 0.1 M PBS was followed by a mixture of 4% paraformaldehyde and 1% glutaraldehyde in 0.1 M phosphate buffer. Brains were removed and postfixed in glutaraldehyde (2.5%, 3 h, 4 °C). Coronal slices at 150 µm thickness were cut using a vibratome (Leica), and the tissue fragments, including CC and SCtx, were picked up under a stereomicroscope. Samples were then postfixed in osmium tetroxide (1%, 1 h, 4 °C), rinsed 3 times with double distilled water for 15 min, and stained en bloc with uranyl acetate (2%, overnight, 4 °C), followed by dehydration in graded ethanol solutions and infiltration with Epon 812 epoxy resin kit (SPI Supplies Division Structure Probe, Inc., West Chester, PA, USA). Polymerized resin blocks were sectioned on an ultramicrotome (Leica UC6). Ultra-thin sections (70–80 nm) were dried on coated copper grids and imaged using a TEM (JEOL 1230) at the Chinese Academy of Sciences.

### 2.8. Intracisternal CSF Tracer Injection and Fluorescence Imaging

Mice in both groups (n = 6/group) were anesthetized using isoflurane (5% for induction, 2% for maintenance) and fixed in a stereotaxic frame. A total of 10 µL of 0.5% CSF tracer, ovalbumin-conjugated Alexa Fluor™ 647 (OA-647) (45 kD, Invitrogen, Waltham, MA, USA) was injected intracisternally at a rate of 1 µL/min with a syringe pump (RWD). Mice were perfusion-fixed 30 min after intracisternal tracer injection. Vibratome slices measuring 100 µm were obtained, mounted with DAPI (Southern Biotech, Uden, The Netherlands), and imaged ex vivo by laser scanning confocal microscopy. To quantify tracer movement into fixed slices, slice images were analyzed in NIH ImageJ software for win64 v.1.8.0_172 as previously described [32]. For each slice, color channels were split and regions of interest (ROIs: whole-slice, Ctx, CC, SCtx) were defined based on the DAPI emission channel. The actual mean fluorescence intensity of the ROIs was obtained after subtracting the mean fluorescence intensity of the background outside the slice. A total of 8–10 slices per mouse were imaged in this manner and mean fluorescence intensity was averaged to generate a single biological replicate.

### 2.9. Statistical Analysis

Data were analyzed using GraphPad Prism 8.0 and were presented as mean ± SD. The number of mice used in each experiment was indicated in the figure legends. Normality was assessed using the Shapiro–Wilk test. Variances were assessed with the Bartlett test for normally distributed data. For comparisons of two groups with normal distributions and equal variances, a two-tailed unpaired *t*-test was used. For the comparison of multiple groups with normal distribution and equal variance, one-way or two-way analysis of variance (ANOVA) with Bonferroni’s posthoc testing was used. *p* < 0.05 was considered statistically significant.

## 3. Results

### 3.1. Characterization of sEVs from Huc MSC

Globular vesicles with diameters around 100 nm were visualized by TEM (Figure 1A). NTA revealed that the size distribution of the vesicles ranged between 50 and 200 nm (Figure 1B). Western blot analysis showed that Alix, TSG101, and CD63 (known EV markers) were enriched in the sEVs pellets (Figure 1C).

### 3.2. MSC-sEVs Were Rapidly Distributed throughout the Brain after Intranasal Administration

To investigate the distribution of MSC-sEVs following intranasal delivery, the fluorescence intensity in the brain was examined over time by NIRF (Figure 2A,B). The treated group exhibited significantly higher radiant efficiency in the brain area than the control group from 0.5 to 6 h post intranasal administration while the radiant efficiency gradually diminished and shrank since 2 h. NIRF imaging showed markedly increased radiant efficiency in the brain (from either dorsal or ventral view) and skull of treated mice at 2 h, with no significant difference observed in the heart, lung, and spleen (Figure 2C,D and Appendix A). Compared to the control group, the liver and kidney of treated mice have modestly increased fluorescence signals, which indicated that a small portion of intranasal sEVs can enter peripheral circulation. Simultaneously, fluorescence of MSC-sEVs was found in all detected brain areas, with a more significant difference in the Str and Hip between the control and the treated group (Figure 2E,F). It was suggested that apart from the OB and Pn, the entry sites of olfactory and trigeminal nerve pathways, MSC-sEVs could rapidly reach many other distant brain regions after intranasal administration.

### 3.3. Intranasal MSC-sEVs Were Incorporated by Multiple Cells in Various Brain Regions

At 2 h post intranasal administration, fluorescence-punctuated signals of MSC-sEVs were observed in microglia, astrocytes, and neurons (Figure 3A,B). The recipient cells were localized in various brain regions, ranging from the OB to Cb, and from the Ctx to SCtx, which was consistent with the subregion-distribution of sEVs as described above. Quantitative analysis revealed no significant difference in MSC-sEVs uptake among these cell types in OB, Ctx, and CC (Figure 3C). Compared with astrocytes, there was a higher percentage of neurons and microglia incorporating MSC-sEVs in the Hip, Pn, and Cb (Figure 3D).

Simultaneously, MSC-sEVs were detected in the dura mater and some were taken up by lymphatic endothelium (Appendix A), suggesting the entry of intranasal MSC-sEVs into the subarachnoid space.

### 3.4. Intranasal MSC-sEVs Were Detected within the PVS

The rapidly and widely central distribution of intranasal MSC-sEVs, as we observed, indicated that CSF flow in PVS may play a key role in sEV distribution throughout the brain. CSF flows along the perivascular pathway and moves into the interstitial space of the brain from PVS, which is outside the vascular wall and externally limited by astrocytic endfeet. In light of this, at 1 h after intranasal treatment with MSC-sEVs, we observed fluorescence of MSC-sEVs within the space surrounded by AQP4-bearing astrocytic endfeet and PDGFR-β positive pericytes in the CC, Str, and Hip (Figure 4A). We took advantage of the fact that GNPs can be resolved under TEM and creatively applied GNP labeling for sEVs. Brain sections were further processed for TEM to capture MSC-sEVs in the microstructure of PVS. MSC-sEVs, recognized as “saucer” like membrane vesicles loaded by GNPs with a diameter of 5 nm, were distributed not only near the basement membrane but also in mural cells (Figure 4B,C).

### 3.5. Intranasal Administration of MSC-sEVs Increased CSF Tracer Influx

The exchange of CSF and ISF is one of the important functions of PVS. Due to the regulatory roles of MSC-sEVs in recipient cells, MSC-sEVs may exert an effect on the exchange of CSF and ISF along perivascular pathways following the uptake by astrocytes and pericytes. To investigate this, at 7 d after intranasal treatment, which was often used as an interval between multiple doses, intracisternal CSF tracer injections were exploited to show the rate of CSF–ISF exchange in the treated and the control groups. To avoid additional effects of arterial pulsation on CSF–ISF exchange, the two groups of brain slices were divided into the anterior (+1.5~−0.8 mm relative to bregma) and posterior (−0.8~−2.5 mm relative to bregma) brain regions, respectively, as determined by the territory of anterior circulation and posterior circulation (Figure 5A) [26,33]. In each brain region, selected fields, including Ctx, CC, and SCtx, were analyzed in 4~5 nonadjacent sections (∼100 μm apart) and averaged per mouse.

Compared with the controls, CSF tracer influx into both the anterior and the posterior brain regions was largely increased (Figure 5B). Subregional analysis showed that increased CSF tracer influx within the anterior brain regions appeared to be more apparent in the SCtx structures, but the effect did not appear in posterior brain regions (Figure 5C,D). These results suggested that intranasal administration of MSC-sEVs increased CSF tracer influx into the brain parenchyma, especially into the SCtx of anterior brain regions where the distribution of MSC-sEVs is more prominent (Figure 2E,F).

## 4. Discussion

As an emerging drug carrier for CNS diseases, sEVs offer unique advantages but also present challenges when moving into the clinic. Common intravenous administration of sEVs results in more peripheral distribution and rapid clearance [34]. To improve the central delivery efficiency of sEVs, intranasal administration is a feasible strategy. It has been identified that there is a rapid enrichment of intranasal sEVs in CNS. Researchers found at 1 h after administration, DiR labeling sEVs given intranasally were detected in the brain, while sEVs administered via a tail vein were mainly distributed in the liver and spleen [35]. The efficiency of nasal delivery of naïve sEVs into the brain was even superior to that of intravenously injected RVG-modified sEVs with brain targeting. Even at 6 h after administration, the fluorescence of sEVs was almost intangible in the brain of the intravenously injected group while strong fluorescence was present in the brain of the intranasally administrated group [36]. Our study also demonstrated the obvious fluorescence of MSC-sEVs in the brain since 0.5 h after intranasal administration, lasting at least 6 h (Figure 2A,B).

Alongside the rapid delivery into the brain, intranasal administration reduces sEVs’ peripheral distribution and clearance. Although some intranasal substances may be absorbed into olfactory blood vessels, or the lower respiratory tract, and even some directly flowing into the digestive tract via the nasopharyngeal tube, the proportion is small, as shown by little fluorescence in the lung, liver, and kidney [20,35,36]. Due to the size distribution and high biocompatibility, sEVs can be transported from the nose to the brain. In our study, we adjusted the position of mice to promote the contact of droplets with the olfactory region serving as the main facilitator for the transfer of molecules to the brain. Permeation enhancer hyaluronidase was applied to enhance sEVs absorption across the olfactory epithelium. Therefore, in line with many other studies, a stronger and more extensive fluorescence of MSC-sEVs was located in the brain than in peripheral organs (Figure 2C,D). However, many studies, including ours, do not examine the exact concentration of sEVs administered in the serum or brain because of a lack of sensitive and reliable detection methods.

Since we had confirmed the fast and centralized distribution of intranasal MSC-sEVs, the layout of MSC-sEVs in the brain exerted further investigation. Within 2 h, the fluorescence of MSC-sEVs was measured in various brain regions and more significant levels were detected in SCtx areas more distant from OB (Figure 2E,F). After being transported into those brain regions, MSC-sEVs could be taken up by glia and neurons residing there (Figure 3). The percentage of cells incorporating sEVs had no significant difference in OB, Ctx, and CC, which may be influenced by the small number of mice (n = 3). This result of regional distribution was similar to that harvested from analysis for the radioactivity of brain sections following intranasal ^125^I labeled MSC-EVs-based therapy for mice of the AD model [37]. In their study, EVs were mostly distributed in the neurons in OB and Ctx. Microglia has also been regarded as one of the dominant recipient cells of MSC-EVs as determined by anti-inflammatory therapy by intranasally delivering EVs [38]. Compared with the cell distribution of MSC-sEVs via other kinds of administration, intranasal delivery does not augment the uptake of naive MSC-sEVs by a specific cell type.

To note, this rapid and extensive distribution after intranasal administration is not sEVs-specific. Kurano et al. depicted the fluorescence of negative liposomes with a diameter around 100 nm, spread throughout the brain 2 h after administration [39]. The distributive pattern of intranasal particles largely depends on their physicochemical properties. Furthermore, the distribution pattern is not exclusive to MSC-sEVs. A similar distribution was observed in intranasal sEVs derived from other cell types, such as dendritic cells [35]. The distribution of sEVs in brain cells can also be influenced by various factors, such as pathological conditions, parental cell types, and modification status. For instance, Kodali et al. reported that at 6 h after intranasal administration, MSC-Exos were robustly incorporated into neurons and microglia throughout the intact and status epilepticus-injured forebrain, with a greater uptake by neurons in regions of brain injury [16]. Intranasal Exos derived from IFNγ-stimulated dendritic cells exhibited the potential to target oligodendrocytes [35]. Additionally, RVG-MSC-Exos were able to target neurons in the substantia nigra at 6 h after intranasal administration [20].

The delivery of MSC-sEVs from the entry point to other distant brain areas after intranasal administration subsequently brought a question: how can this kinetic characteristic be achieved? Researchers have suggested that the perivascular pathway was mostly important to the rapid and widespread distribution of macromolecules in CNS following intranasal delivery [25,26,40], confirmed by the fact that fluorescence-labeled dextrans (3 or 10 kDa), peptide (<10 kDa), or IgG (150 kDa) was rapidly distributed to cerebral PVS. Lochhead et al. have shown that the concentration of IgG in PVS of major cerebral arteries at 30 min was higher than that in corresponding brain regions, but consistent with that in the OB and trigeminal nerves [26]. Thus, we hypothesized that apart from perineuronal diffusion, intranasal sEVs could be transported along the perivascular pathway, followed by the influx into PVS at the capillary level, and reach a widespread distribution into nearby cells and brain parenchyma with the CSF–ISF exchange (Figure 6).

To identify our hypothesis, we observed the colocalization of PVS and MSC-sEVs labeled by a fluorescence dye and the GNP, respectively, at an earlier time point. CSF–ISF exchange is thought to be driven by arterial pulsation and mediated by AQP4 channels on astrocytic endfeet [27]. Pericytes regulate AQP-4 polarization (AQP-4 expression focusing on astrocytic endfeet surrounding blood vessels), which has a profound impact on bidirectional fluid exchange [41]. Given the uptake of intranasal MSC-sEVs by pericytes or astrocytes surrounding PVS, MSC-sEVs may affect the CSF–ISF exchange along perivascular pathways, which will further support our hypothesis. Our results showed that the CSF–ISF exchange was promoted in the treated group, especially in the SCtx of anterior brain areas where MSC-sEVs were more abundant. Taken together, our findings suggested that CSF–ISF exchange in the PVS may well be the momentum to widespread central delivery of intranasal MSC-sEVs, and MSC-sEVs could have a positive effect on the exchange probably due to the incorporation by perivascular cells.

The intranasal sEVs-based drug delivery provides a promising way to enhance the safety and efficacy of drugs for CNS diseases. However, there are some limitations and caveats in evaluating preclinical results. For example, the nasal anatomy of humans is different from that of rodents. The olfactory epithelium serves as the main facilitator for the transfer of molecules to the brain and covers only 3% of the nasal cavity in humans, compared to 50% in rodents [42]. In addition, mild anesthesia may lower the velocity of inhaled droplets and increase contact with the nasal mucosa [13]. The permeation enhancer applied would promote the transportation from the nasal mucosa to OB. Therefore, animal experiments are more likely to show greater advantages of intranasal sEVs, but the effects may be weak in clinical practice. Data from the large animals will be needed and the formulation of sEVs loading drugs applying to the clinic warrants to be improved.

## 5. Conclusions

In this study, we confirmed the effective delivery and retention of MSC-sEVs in CNS after intranasal administration. We also demonstrated the rapid and extensive distribution of MSC-sEVs in various brain regions and different cell types, particularly in the subcortex. Our findings suggested that the PVS is involved in the rapid and widespread central delivery of intranasal MSC-sEVs.

## Figures and Tables

**Figure 1 pharmaceutics-15-02578-f001:**
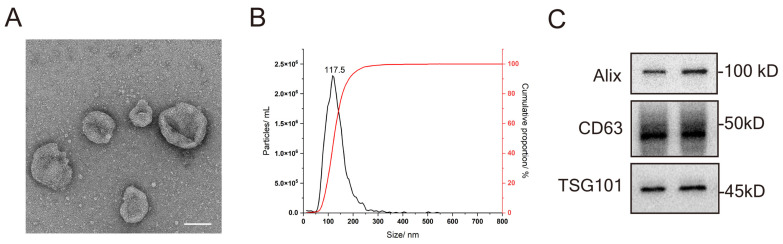
Characterization of sEVs from huc MSC. (**A**) Representative morphology of huc MSC-sEVs under TEM (scale bar 100 nm). (**B**) NTA analysis showing the size range of sEVs. (**C**) Western blot analysis showing classic EV proteins, including ALIX, TSG101, and CD63.

**Figure 2 pharmaceutics-15-02578-f002:**
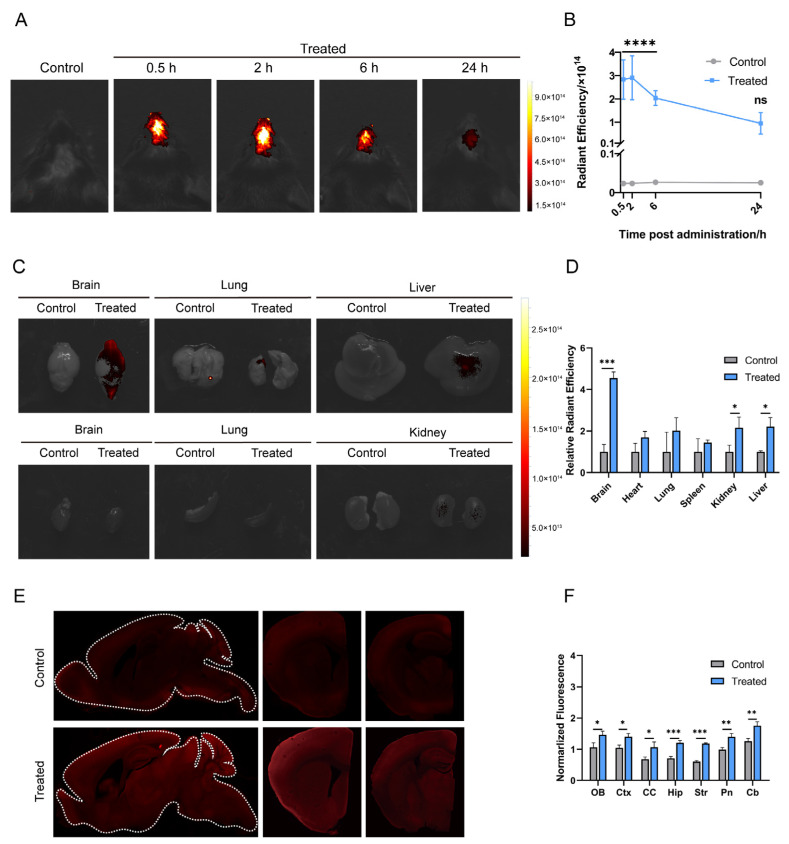
Biodistribution of MSC-sEVs following intranasal delivery. (**A**) Representative images of Cy7-MSC-sEVs fluorescence in the brain at 0.5, 2, 6, and 24 h after intranasal delivery. (**B**) Quantification of radiant efficiency of Cy7-MSC-sEVs administered. The treated group exhibited more significant radiant efficiency in the brain area at 0.5, 2, and 6 h while there was no significant difference at 24 h. (**C**) Representative images of Cy7-MSC-sEVs fluorescence in different organs at 2 h after delivery. (**D**) Quantification of relative radiant efficiency of Cy7-MSC-sEVs administered. (**E**) Representative images of Cy3-MSC-sEVs fluorescence in brain slice at 2 h after administration. (**F**) Quantification of normalized fluorescence intensity of Cy3-MSC-sEVs in all brain regions. * *p* < 0.05, ** *p* < 0.01, *** *p* < 0.001, **** *p* < 0.0001, ns = not significance. Data are presented as mean  ±  SD, n = 4/group, (**B**) two-way ANOVA with Bonferroni’s posthoc testing, and (**D**,**E**) two-tailed unpaired *t*-test. OB: olfactory bulb, Ctx: cortex, CC: corpus callosum, Str: striatum, Hip: hippocampus, Pn: pons, and Cb: cerebellum.

**Figure 3 pharmaceutics-15-02578-f003:**
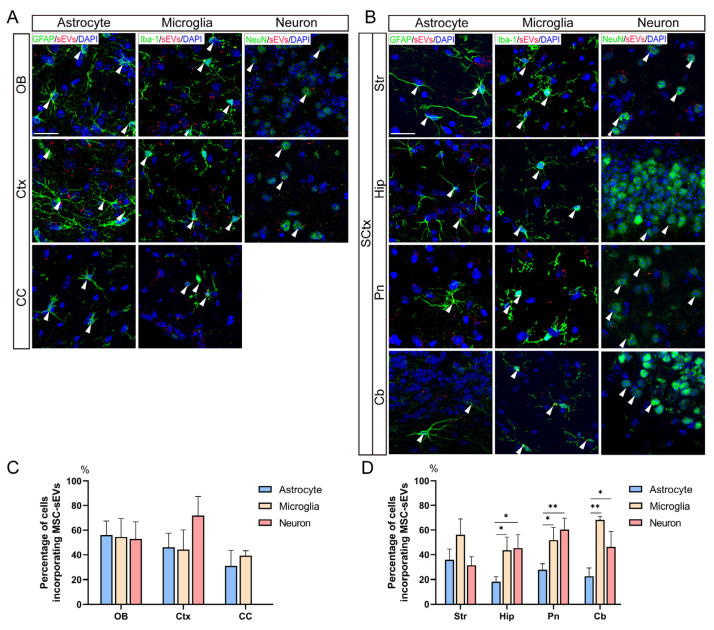
Uptake of MSC-sEVs by multiple cells in various brain regions. Representative images showing the uptake of Cy3-MSC-sEVs (red/yellow dots) by microglia, astrocytes, and neurons in the OB, Ctx, CC (**A**), and other subcortex structure (SCtx), including Hip, Str, Pn, and Cb (**B**) (scale bar 25 µm). The white arrowheads indicate the presence of Cy3-MSC-sEVs within the cytoplasm of cells. Quantification of the percentage of cells containing sEVs in the OB, Ctx, CC (**C**), and SCtx (**D**). * *p* < 0.05, ** *p* < 0.01. Data are presented as mean  ±  SD, n = 3/group, (**C**,**D**) one-way ANOVA with Bonferroni’s posthoc testing. OB: olfactory bulb, Ctx: cortex, CC: corpus callosum, Str: striatum, Hip: hippocampus, Pn: pons, and Cb: cerebellum.

**Figure 4 pharmaceutics-15-02578-f004:**
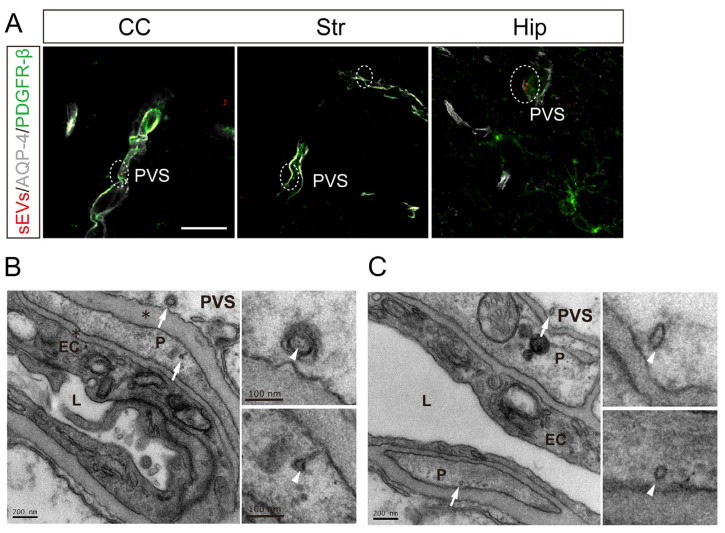
Localization of intranasal MSC-sEVs concerning the PVS. (**A**) Representative image of Cy3-MSC-sEVs captured in the PVS located in the CC, Str, and Hip (scale bar 10 µm). (**B**,**C**) Representative images of TEM showing the location of GNP-MSC-sEVs within the PVS. The white arrows indicated GNP-MSC-sEVs near the basement membrane and in pericytes (**left**) (scale bar 200 nm), and the white arrowheads showed magnified views of loaded GNPs (**right**). CC: corpus callosum, Str: striatum, Hip: Hippocampus, PVS: perivascular space, EC: endothelial cells, P: pericytes, L: capillary lumen, and *: basement membrane.

**Figure 5 pharmaceutics-15-02578-f005:**
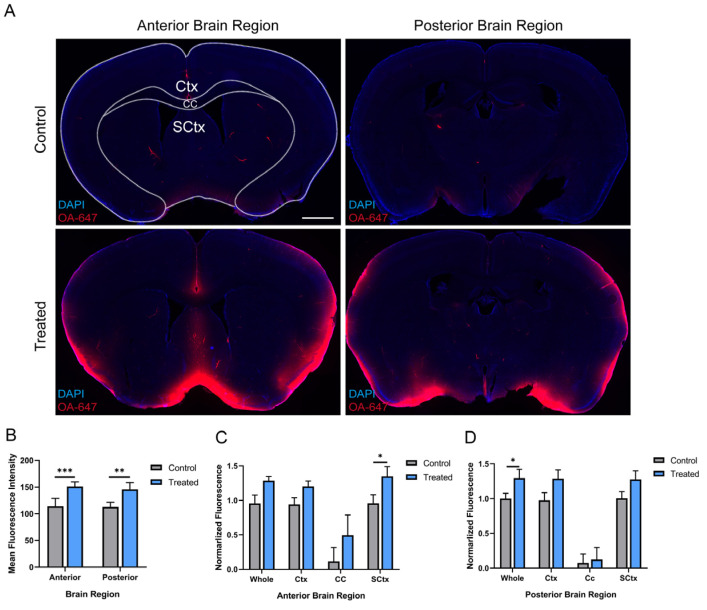
Intranasal administration of MSC-sEVs increased CSF tracer influx. (**A**) Representative image showing the distribution of CSF tracer OA-647 30 min after intracisternal injection in the control and treated group (scale bar 1 mm). (**B**,**D**) Quantification of CSF tracer influx into all brain regions (**B**), anterior (**C**), and posterior (**D**) brain regions. * *p* < 0.05, ** *p* < 0.01, *** *p* < 0.001. Data are presented as mean  ±  SD, n = 6/group, (**B**,**D**) two-way ANOVA with Bonferroni’s posthoc testing. Ctx: cortex, CC: corpus callosum, and SCtx: subcortex structure.

**Figure 6 pharmaceutics-15-02578-f006:**
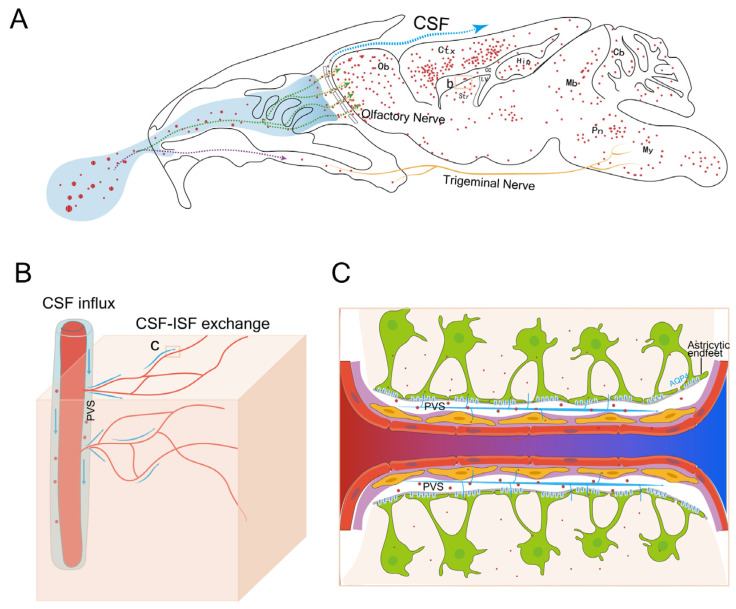
Schematic of a distributive pattern of MSC-sEVs following intranasal administration. (**A**) Intranasal MSC-sEVs (red dots) can reach the OB or Pn along the axons of the olfactory nerve and trigeminal nerves and have the potential to enter the olfactory subarachnoid space full of CSF. Subsequent distribution of MSC-sEVs in the brain may be mediated by the perineuronal pathway, which may result in local diffusion of MSC-sEVs near OB and Pn and perivascular pathway by which MSC-sEVs can be rapidly transported to various brain regions. (**B**) Intranasal MSC-sEVs with CSF influx can reach every level of PVS where CSF–ISF exchange occurs. (**C**) Through the exchange within pericapillary space, MSC-sEVs can be incorporated into nearby pericytes or astrocytes and distributed into the brain parenchyma, which is facilitated by the polarization of AQP 4 toward the PVS. OB: olfactory bulb, Ctx: cortex, CC: corpus callosum, Str: striatum, Hip: hippocampus, Mb: midbrain, Pn: pons, My: medulla, Cb: cerebellum, PVS: perivascular space, CSF: cerebrospinal fluid, ISF: interstitial fluid. Green and purple arrows showed the olfactory and trigeminal pathways respectively while blue arrows depicted the perivascular pathway.

## Data Availability

Data will be provided upon reasonable request.

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
