# Peer review of "Rapid and Widespread Distribution of Intranasal Small Extracellular Vesicles Derived from Mesenchymal Stem Cells throughout the Brain Potentially via the Perivascular Pathway"

_pharmaceutics, 2023, doi:10.3390/pharmaceutics15112578_

Round 1
Reviewer 1 Report
Comments and Suggestions for Authors
Introduction:
- Provides necessary background on exosomes, limitations of IV delivery, and advantages of intranasal administration. Concise and focused.
- Paragraph 2 could improve flow by reordering some sentences. Move line 27-29 later after introducing intranasal administration.
- More citations could be added for statements on BBB permeability and distribution of IV exosomes.
- The objectives could be stated more clearly at the end of the intro.
Methods:
- Overall, the methods are well-described, with sufficient detail provided.
- For Western blots, include details on SDS-PAGE parameters, transfer apparatus, blocking and antibody incubation conditions, washing buffers, imaging systems, etc.
- Clarify concentrations and volumes of reagents used for exosome labeling reactions.
- Specify angle of head tilt during intranasal administration.
- State thickness of brain slices imaged.
- Provide paraformaldehyde/glutaraldehyde percentages for postfixation.
- Include number, age, sex of animals used.
- Describe specific statistical tests performed for data analyses.
- Include adherence to the MISEV2018 guidelines and mention any deviations from them and how it may or may not affect the final results. You may find more information of the MISEV guidelines from the paper below:
- https://www.mdpi.com/2218-273X/13/5/839#:~:text=PDEVs%20can%20also%20be%20used,healthcare%20products%20in%20the%20future
Results:
- The results are logically structured and presented clearly in text/figures.
- Figure legends could be expanded to define abbreviations for brain regions.
- In figure 4, increase font size for labels indicating perivascular space anatomy.
- For Figure 5, include information on slice selection and quantification methods.
- Statistical values and variation could be reported more consistently throughout text (e.g. always include N, error bars).
Discussion:
- The discussion is organized well but is overly long. It could be streamlined to focus on key findings.
- Some sentences are repetitive or redundant. For example, line 306-309 essentially repeats earlier points.
- The section on cell targeting is interesting but disrupts the focus on distribution mechanisms. Should be moved to later in the Discussion.
- More attention could be given to limitations and caveats, including anesthesia effects, small animal models, exosome quantification, etc.
Overall Evaluation:
- This is a well-executed, reasonably comprehensive study providing valuable insights into intranasal exosome delivery to the brain.
- With relatively minor revision to improve clarity in some areas, enhance details in methods, refine presentation of results, condense discussion and highlight limitations more clearly, this paper would represent a significant contribution to the literature in this emerging field.
- I recommend this manuscript for publication in Journal of Pharmaceutics pending minor revisions. The findings have important implications for developing intranasal exosome therapeutics.
Reviewer 2 Report
Comments and Suggestions for Authors
Specific comments
Due to the overlapping size range and the lack of specific markers, the current extracellular vesicle (EV) preparations including exosome preparation are highly heterogeneous with undetermined purity and/or undefined biogenesis origin. As such, authors are recommended to refer to the position papers of the International Society for Extracellular Vesicles (ISEV) for guidelines on the nomenclature, isolation, and characterization. Authors should therefore use the term EVs or small EVs instead of exosomes.
J Extracell Vesicles. 2019 Apr 29;8(1):1609206.
J Extracell Vesicles. 2018 Nov 23;7(1):1535750.
Authors should improve the writing. Some sentences appear strange. For instance, “In this study, we more concentrated on the distributive pattern of Exos after brain enrichment.”
Figure 3 – Higher magnification and better resolution images should be provided. It is not possible to see the red/yellow dots using the current images provided. Author should also provide (semi) quantitative data to substantiate the results on uptake of MSC-EVs by the different cell types (microglia, astrocytes, and neurons) in the various brain regions, and discuss if there is preferential uptake of the MSC-EVs by a certain cell type.
Figure 5B, C and D – the statistical annotation between the “Control” and “Treated” should be done differently. It is currently not clear what the statistical annotation meant. All statistical annotations should be made only within the graphs.
Figure 5 – Can the authors explain why the CSF tracer influx was more prominent in the periphery of the cortex?
Comments on the Quality of English LanguageThe English writing should be improved.
Reviewer 3 Report
Comments and Suggestions for Authors
The manuscript “Rapid and widespread distribution of intranasal exosomes derived from mesenchymal stem cells throughout the brain potentially via perivascular pathway” describes a possible approach for a drug delivery system to the central nervous system based on exosomes.
In my opinion, the manuscript has several minor issues that must be addressed before it can be considered for publication.
The authors should clarify for what kind of drugs this system is feasible and how they will be loaded onto or into exosomes and why the exosomes are highly accumulated in the brain and not in the respiratory system instead.
Comments on the Quality of English LanguageMinor editing of English language required
Reviewer 4 Report
Comments and Suggestions for Authors
The authors examined the distribution of exosomes from mesenchymal stem cells in the brain after intranasal administration of the exosomes, as intranasal administration is a promising strategy to enhance the delivery of exosome-based drug delivery system to CNS. They mentioned that the exosomes were rapidly distributed throughout the brain via the perivascular pathway. They mentioned that the exosomes were rapidly distributed throughout the brain via the perivascular pathway and suggested that the intranasal route to deliver the exosomes for CNS therapy is very important. The aim and concept of this study are reasonable; however, this work still needs many revisions to be accepted.
1. When exosomes were labeled with fluorescent dyes and GNPs, great attention should be paid to unconjugated particles. In the text, the authors mentioned that ultracentrifugation was performed for the removal of unconjugated particles. In general, ultracentrifugation is not precise for the removal because unconjugated particles are also precipitated, as shown in many previous studies. In addition, I wonder if 30% sucrose cushion is successful for the separation in this study condition. The authors should show evidence for the success of the separation. Further, the authors should compare the effects of particles such as dyes and GNPs in the presence or absence of exosomes. The current results cannot deny the influence of unconjugated particles.
2. In Fig.4, the authors mentioned that the white arrows indicate GNP-labeled Exos, but is it possible that smaller ones are unconjugated particles?
3. Is this the exosome-specific phenomenon after intranasal administration of exosomes? How about liposomes and microvesicles? In addition, although the authors used exosomes from mesenchymal stem cells in this study, are there differences among cell-types?
4. In general, exosome uptake differs among cell types as the authors know. In this study, the authors should show the details.
5. I am not sure about the authors' interpretation of Fig.5. Fig.5 showed that the permeability of the CSF tracer was altered by the intranasal administration of something. The authors must provide evidence that this is an effect of exosomes. The result could also be interpreted because of the breakdown of the barrier-like structure by intranasal administration of something. In addition, why do not the authors show the results at 2h? In Fig.2B, the authors showed significant change at 2h.
6. I want the authors to confirm the statistical analysis in this study again. For examples, in Fig.2B, I cannot understand the significance at 2h and ns (not significance?) at 24h. How about 0.5h and 6h?
7. In Fig.3, I cannot find red/yellow dots partly. The authors need to confirm the resolution of images.
8. Supplement data of legend is insufficient. For example, an explanation of D is not found.
Comments on the Quality of English LanguageIn general, the manuscript is well written.
Round 2
Reviewer 2 Report
Comments and Suggestions for Authors
Most of the reviewers' comments have been addressed.
Comments on the Quality of English LanguageThe quality of English can be improved.
Reviewer 4 Report
Comments and Suggestions for Authors
The authors well-addressed all my previous comments. The manuscript was significantly improved thus it can be accepted for publication after the editorial check.